# Growing Trial of Gilthead Sea Bream (*Sparus aurata*) Juveniles Fed on Chironomid Meal as a Partial Substitution for Fish Meal

**DOI:** 10.3390/ani9040144

**Published:** 2019-04-03

**Authors:** Alessandra Roncarati, Roberto Cappuccinelli, Marina C.T. Meligrana, Roberto Anedda, Sergio Uzzau, Paolo Melotti

**Affiliations:** 1School of Bioscience and Veterinary Medicine, University of Camerino, 62032 Camerino, Italy; marina.meligrana@unicam.it (M.C.T.M.); paolo.melotti@unicam.it (P.M.); 2Porto Conte Ricerche, Edificio 10 del Parco Scientifico e Tecnologico della Sardegna, 09010 Pula, Italy; cappuccinelli@portocontericerche.it (R.C.); anedda@portocontericerche.it (R.A.); uzzau@portocontericerche.it (S.U.)

**Keywords:** gilthead sea bream, chironomid, insect meal, feed, growth performance

## Abstract

**Simple Summary:**

In fish feeding, importance has been placed on the search for alternative ingredients to fish meal and fish oil due to the decline in fishery supplies and high fluctuations in the price of aquatic ingredients. One of the most promising alternative feedstuffs to date is insects as they are considered to be a sustainable source of amino acids and other essential nutrients. In this study, we evaluated the growth performances of gilthead sea bream that were fed two diets containing different amounts of insect meal, composed of chironomids at the larval stage, in order to reduce the protein source provided by fish meal. Chironomids were collected from aquatic environments, processed, analyzed, and included in these two feeds. We ascertained that the two feeds containing chironomid meal were well accepted. Both the replacements in the diets resulted in suitable growth performances and were not significantly different from the growth that resulted from the fish fed the control diet. We suggest that it is possible to harvest chironomids when the maximum concentration of larvae is found in the aquatic environment, or the other alternative is to culture them in ponds or natural basins. In this way, we can add the chironomid species to the list of insects that can be used for feed production in aquaculture.

**Abstract:**

Insect meal derived from chironomid larvae and collected from aquatic environments was included in the feed of gilthead sea bream juveniles (75 ± 1.1 g) in a growth trial of 90 days. Three feeds, which were namely one control (L1) and two experimental diets (L2, L3), were analyzed and formulated as isonitrogenous (45%) and isolipidic (13%). In L1, the protein source was mainly soybean meal (32%), followed by fish meal (20%), wheat meal (20%), gluten corn (17%), and hemoglobin (11%). In L2, the proportion of soybean meal was increased (33.5%), followed by gluten corn (21%), wheat meal (14%), and hemoglobin (11%), whereas the fish meal source was reduced (15%) due to the inclusion of chironomids (5%). In L3, the proportion of fish meal was further reduced (8%) and that of chironomid meal was increased to 10% of the protein source. The L2 and L3 groups showed similar growth performances with respect to the L1 group. The feed conversion rate was favorable in all the groups, ranging from 1.18 (L1) to 1.22 (L3). Survival rates varied from 93.62% (L3) to 94.31% (L1). Feed palatability showed similar results for all diets. Although the inclusion of chironomid meal was used in small quantities, our results suggest a significant advantage in replacing 50% of the fish meal with the chironomid meal for growing gilthead sea bream fishes.

## 1. Introduction

In the last few years, insect meal has become one of the most studied sources of protein feedstuffs as an alternative to fish meal for aquaculture feed. The larvae, pupae, and adults of various insects are increasingly assayed to determine which invertebrates can be used in variable processed meals as part of feed formulations for aquaculture species. The Food and Agriculture Organisation (FAO) recommends using insects since they allow for sustainable productions, especially considering the high nutritional content of many edible insects and their minimal ecological impact [1]. Some species have been shown to be compatible with animal feed in aquaculture, as contemplated by the Reg. UE 2017/893, and exhibit great potential as feed ingredients due to their good nutritional quality [2,3].

Insects with aquatic larval stages have been less frequently studied than mealworms and flies. Some studies have investigated aquatic invertebrates as they could represent an important source of nutrients for both humans and livestock. Among lake flies, chironomids are common in aquatic environments and are found at different latitudes [4,5]. Furthermore, they are the prey of different fish species. There are different chironomid species, including freshwater, euryhaline, and marine species. Although they have seasonal availability, these insects can reach a very high density in their natural habitat, creating swarms of flying insects in summer [6]. Insects are naturally consumed as a feed source in the natural water environment by salmonids and carnivorous fish species [7]. Eggs deposited on the water surface after each swarming event soon develop into larvae. In these situations, the chironomid larvae can be captured in high quantities and employed after dehydration for ornamental fish feeding [8]. The freshwater species can be used as a natural food source to improve growth in fish juveniles since they show a more favorable proportion of long-chain polyunsaturated (PUFA) fatty acids compared to terrestrial insects [9].

In the last few years, chironomids have often been overabundant in certain seasons of the year and caused trouble in different areas. In recent times, flights in the airports close to the North Adriatic coast have been blocked due to the blooms of chironomid flies. In the Trasimeno Lake, the densities of chironomids are very high in the summer season and much effort is spent in controlling or reducing them [10]. Recently, a study was performed on chironomid larvae in basins and channels in which a high quantity of larvae was captured and dehydrated. The chemical composition analyses showed good nutritional profiles in the chironomid larvae meals, making them suitable for fish feeding [11]. Continuing the studies on the possible inclusion of chironomid in feeds for aquaculture species, a growth trial was performed to assess the effects of the meal obtained from chironomid species collected from an aquatic environment on gilthead sea bream (*Sparus aurata* L.). In order to achieve this, samples of midges at the larval stage were converted into meal and first analyzed from a qualitative point of view. After this, the chironomid meal was included in the feedstuffs for gilthead sea bream juveniles.

## 2. Materials and Methods

### 2.1. Chironomid Sampling, Meal Processing, Feed Preparation, and Chemical Analyses

During the spring and summer of 2017, around 100 kg of chironomid larvae was collected using nylon net traps from ponds and aquatic environments, which were located between the Emilia Romagna and Marche regions. The larvae were washed and kept at a temperature of 4 °C for 24 h. Subsequently, the larvae were pooled and dried at a temperature of 70 °C before being milled. The meal samples were analyzed for their proximate composition (moisture, protein, lipid, and ash content) as well as their amino acid and fatty acid profiles, whereas the other lots were stored, waiting to be included in the diets. The same analyses were performed on the fish meal that was going to be included in the experimental feeds (Table 1 and Table 2).

Three feeds, which were namely one control (L1) and two experimental diets (L2, L3), were formulated to have a similar energy content and were isonitrogenous (around 45%) and isolipidic (around 13%). In the L1 feed, the chironomid meal was absent and the protein source was mainly represented by soybean meal, as reported in Table 3. In L2, the proportion of soybean meal as a protein source was increased (33.5%), whereas the source of fish meal was reduced (15%) due to the inclusion of chironomid meal (5%). In the L3 feed, the fish meal was further reduced (8%) and the chironomid meal was increased to 10% of the protein source (Table 3).

For manufacturing the three different diets, the meal of each one was mixed with the other ingredients. The dough of each three was pelleted by pressing it through a sieve (2.2 mm mesh) in a small-scale laboratory pellet mill (Zhengzhou Pasen Machinery Co., Ltd., Zhengzhou, China). After this, the pellets were dried in a thermostatic drying oven at 40 °C until the moisture level decreased below 8% before being stored at 4 °C in black bags.

The chemical analyses of the three samples of each feed were performed according to the procedure outlined by the Association of Official Analytical Chemists [12]. The total lipid content was determined using the procedure described by Folch et al. [13]. The essential amino acids in the three feeds were determined by acid hydrolysis (6 N HCl for 24 h at 110 °C), which was followed by ion exchange chromatography utilizing an amino acid analyzer (L-8800 Auto-analyzer, HITACHI, Tokyo, Japan). After determining the total lipid content, the fatty acids were converted to methyl esters following the method described by Christopherson and Glass [14]. The separation of fatty acids was carried out using a GC 3800 gas chromatograph (Varian Strumentazione, Cernusco sul Naviglio, Italy) with a WP-4 Shimadzu integration system (Shimadzu Corporation, Tokyo, Japan), which was equipped with a Supelco SPTM-2340 capillary column (30 m × 0.25 mm internal diameter; 0.25 μm film thickness; Supelco, Bellefonte, PA, USA) and a flame ionization detector.

### 2.2. Fish and Growth Trial

For the trial, 460 sea bream fingerlings that weighed 75 ± 1.1 g were randomly distributed at a density of 8 kg/m^3^ in nine 2 m^3^ indoor tanks, which were supplied by three separate recirculating aquaculture systems at 21.0 ± 0.5 °C. To avoid possible differences associated with different recirculating water systems, each feed was administered in one of the three tanks of each water circuit. Each feed was assigned to three tanks and distributed twice a day ad libitum over 6 days per week. Any excess feed was removed after 15 min using a siphon hose system. The feeding trial lasted 90 days and the fish were weighed individually at the beginning of the trial and subsequently on a monthly basis. The palatability of the feeds was assayed according to the formula: (ingested feed/administered feed) × 100.

### 2.3. Morpho-Biometric Parameters and Indices

Body weight was measured using an electronic scale (Ohaus Adventurer SL Precision Balance, Mod: AS8100) and total body length was determined using a metric scale. At the end of the feeding experiment, the following indices were also determined (50 fish/tank/feed): condition index (KI) = ((100 × fish weight)/fish length^3^), viscerosomatic index (VSI) = (viscera weight/whole body weight) × 100, perivisceral fat index (PFI) = (perivisceral fat/body weight) × 100, and hepatosomatic index (HSI) = (liver weight/body weight) × 100. In order to measure the PFI and the VSI, the fat adherent to the digestive tract was accurately separated and individually weighed.

### 2.4. Water Quality

During the trial, the main water physicochemical parameters (temperature, dissolved oxygen, and pH) of the three recirculating systems were recorded on a weekly basis. The total ammonia nitrogen (TAN), nitrites (NO_2_), and nitrates (NO^3^) were analyzed following the American Water Works Association and Water Pollution Control Federation of American Public Health Association (APHA) standard methods [15].

### 2.5. Statistical Analysis

The data were subjected to one-way analysis of variance (ANOVA) using the General Model procedure of SAS [16]. Differences were considered to be significant if *p* < 0.05 and the means were compared using the Student–Newman–Keuls (SNK) test.

## 3. Results

Throughout the trial, the main water physicochemical parameters were within the range that is considered to be optimal for the species [17]: pH > 7; dissolved oxygen > 6 mg/L; salinity 37 ± 1 ppt; ammonia, nitrogen, and nitrites below the detection limit; and nitrates < 60 mg/L.

The chemical composition and essential amino acid content of both the chironomid and the fish meal are reported in Table 1, while the fatty acid profile is shown in Table 2. The chemical composition of the chironomid meal showed a lower protein and ash content and a higher lipid content with respect to the fish meal. The amino acid composition of the chironomid meal had a higher content of Valine (VAL), Threonine (THR), and Phenylalanine (PHE) with respect to the fish meal whereas Arginine (ARG), Histidine (HIS), Isoleucine (ISO), Leucine (LEU), Lysine (LYS), and Methionine (MET) were higher in the fish meal. With respect to the fatty acid profile, the saturated fatty acids (SFA) fraction in the chironomid meal was lower (34.75%) than in the fish meal (43.79%); the monounsaturated fatty acids (MUFA) fraction was higher in the chironomid meal (37.83%) than in the fish meal (30.2%). The PUFAs ɷ6 (21.86%) were higher in the chironomid meal compared to the fish meal (4.1%) whereas the PUFAs ɷ3 were higher in the fish meal (21.91%) with respect to the chironomid meal (5.56%).

The ingredients, chemical composition, energy, and amino acid profile of the three diets are reported in Table 3, and the fatty acid profile is shown in Table 4. The chemical composition of the three diets was very similar. With regard to the amino acid profile, PHE, THR, Tryptophan (TRP) and VAL were higher in L3 compared to L1 and L2.

Concerning the fatty acid profile, differences were observed in the MUFAs, which were higher in L3 (27.87%) and L2 (26.6%) in comparison with L1 (24.16%). The total amount of PUFAs ɷ6 was higher in L3 (14.42%) and L2 (11.41%) compared to L1 (7.04%), while the total amount of PUFAs ɷ3 was lower in L3 (13.72%) compared with L2 (17.06%) and L1 (23.99%) due to eicosapentaenoic acid (EPA) variations in L1 (10.16%), L2 (7.04%), and L3 (6.26%) as well as docosaesaenoic acid (DHA) variations in L1 (9.75%), L2 (5.86%), and L3 (3.1%).

Growth performances, morpho-biometric parameters, and somatic indices of gilthead sea bream juveniles fed the three diets are reported in Table 5. The L2 and L3 groups showed similar growth performances with respect to the L1 group as there were no differences in the final mean body weights and specific growth rates. The final mean body weights exhibited good results with an increase of 127–132% after 90 days of trial. The feed conversion rate was favorable in all the groups and ranged from 1.18 (L1) to 1.22 (L3). The survival rate was high in all three groups with no notable differences as it ranged from 94.31 (L1) to 93.62 (L3). Feed palatability showed similar results in all three diets. With regard to the morpho-biometric parameters and somatic indices, no significant differences were observed among the groups in KI, PFI, VSI, and HSI.

## 4. Discussion

In this trial, a diet formulation similar to that adopted by the aquafeed companies was employed, in which there is a lower amount of fish meal that is compensated by protein sources of vegetable and terrestrial origin. Soybean was used as the main protein source (32–33%), followed by gluten and wheat meals, which are considered more sustainable than fish meal. Hemoglobin was also included to guarantee the presence of high digestible proteins due to its high lysine and leucine contents [18]. In this way, we obtained an appropriate combination of different protein sources as reported by different studies, which had advantages in terms of growth. This was attributed to the synergic effect of different proteins [19,20,21], which was able to mask the unpalatable substances present in feed ingredients. The inclusion of chironomid meal in the feeds represented 5% and 10% of the protein source of the two experimental feeds, respectively. In this way, the replacement of the fish meal reached 25% and 50%, respectively, in the two diets that included chironomid meal. In both groups, when the level of chironomid insect inclusion reached 10% (L3) or was limited at 5% (L2) for replacing fish meal, the gilthead sea bream showed satisfactory growth performances that were similar to the control group (L1).

Many studies have focused on the search for the most suitable insect species to replace the more expensive and unsustainable fish meal in aquaculture diets and the results are generally encouraging. In juvenile Jian carp (*Cyprinus carpio*) that were fed diets with different levels of substitution of soybean oil with black soldier flies, no differences in growth performances were reported [22]. In European sea bass (*Dicentrarchus labrax*) juveniles, the inclusion of *Tenebrio molitor* to replace 50% of the fish meal resulted in a whole-body crude protein and ether extract that was not significantly influenced by the use of insect meal [23]. Other authors [24] recently observed improved growth performances with respect to a control diet when gilthead sea bream juveniles were fed a formulation that included *Tenebrio molitor*. These authors employed sea breams that weighted 105.1 g, which reached market size after 163 days. They attributed these positive results to the inclusion of a moderate percentage of insect meal and the effect of chitin on protein digestibility since it confers a higher protein binding capacity. In our study, the market size was not reached because smaller juveniles (75 g) were employed and the trial lasted only 3 months. However, the fishes exhibited good growth performances. With regard to the somatic indices, the similarity of the results indicated that no adverse effect was observed and the assimilation of the L2 and L3 diets was efficient.

The chironomid meal had a similar composition to the chironomids grown in controlled conditions on pure cultures of microalgae (Habib et al. [25]). However, the preparation used in the present study had higher protein and lower lipid contents relative to the chironomids grown in an algal culture by Habib et al. [25]. In our case, the origin of chironomids from earth-bottom substrates, naturally colonized by microalgae and zooplankton, improved the amino acid profile of chironomids. The levels of tryptophan, leucine, lysine, phenylalanine, threonine, and valine were similar to fish meal and placed this species of insects among those with an adequate composition among the order Diptera, as reported in a review regarding the body composition of several insects compared with the main sources for animal feeding [26]. The levels of these amino acids are different from almost all other insect meals, which are considered to be low in lysine, histidine, and tryptophan for fish [27,28], whereas *Hermetia illucens* meal is also limited in terms of threonine and sulfur amino acids [2]. In the chironomid meal used in the present study, the essential amino acid profile was consistent with the estimated requirements for gilthead sea bream [29].

Regarding the fatty acid profile, according to Raksakantong et al. [30], insects characterized by an aquatic life cycle, such as chironomids, are different from terricolous insects because they contain long-chain PUFAs due to their diet and enzymatic activity, which may increase the synthesis of long-chain PUFAs. In the literature [8,26], chironomid insects have been reported to have a significant content of EPA (ranging between 3.4% and 11.6% of total fatty acids), which is higher than that recorded in our study, although the results of DHA are consistent with that found in the literature (0.5% compared to 0.1 and 0.2%). In juvenile lake trout [31], after a 14-week period, better growth was exhibited in fish receiving chironomids compared with those fed different invertebrates, such as copepods and *Mysis*, despite the low long-chain essential fatty acids.

Concerning diet acceptability, the addition of chironomid meal did not suppress growth performance, which is presumably due to the fact that chironomids represent a natural prey for many freshwater and euryhaline fish species. As such, chironomid meal may have functioned as a dietary attractant in the two experimental feeds.

The chironomids used in the current study were harvested from wild freshwater environments. The FAO recommends that the insects should be obtained through production activities in order to guarantee the security of the product and to avoid threatening existing populations [6,32]. However, as mentioned previously, chironomids are often overabundant in certain seasons of the year and can be a nuisance in many areas. Thus, chironomids could be harvested when the maximum concentration of larvae is found in the aquatic environment. It is also possible to trigger their production and induce a midge-larvae abundance of chironomids using available techniques, such as the reported channel catfish fingerlings reared in plastic-lined and earthen basins [33]. After this, the processing of meal could be performed, and this meal could be stored, waiting to be used. Although the chironomid meal was used in small quantities, our results suggest that there is a significant advantage in replacing 50% of the fish meal with chironomid meal during the growth of gilthead sea bream.

Further studies are required to fully optimize the employment of the meal obtained by this specific insect in order to propose it as an insect species to add to the list of insects employed for feed production in aquaculture.

## Figures and Tables

**Table 1 animals-09-00144-t001:** Chemical composition (g/kg) and essential amino acid (g/kg dry matter) content of chironomid meal (CM) and fish meal (FM).

**Nutrient Component**	**CM**	**FM**
Dry matter	910.4	920.0
Crude protein	585.6	710.0
Crude lipid	145.0	124.0
Ash	107.9	167.0
**Essential Amino Acid Content**	**CM**	**FM**
Arginine (ARG)	43.0	49.5
Histidine (HIS)	19.0	21.0
Isoleucine (ISO)	22.0	32.0
Leucine (LEU)	58.7	62.0
Lysine (LYS)	51.6	56.0
Methionine (MET)	22.8	27.0
Phenylalanine (PHE)	31.6	28.0
Threonine (THR)	37.4	32.0
Tryptophan (TRP)	7.9	8.3
Valine (VAL)	38.6	33.0

**Table 2 animals-09-00144-t002:** Fatty acid profile (% of total fatty acids) of CM and FM used in the trial.

Fatty Acid	CM	FM
14:0	1.99	10.6
15:0	0.94	0.49
16:0	19.92	26.64
17:0	1.74	0.98
18:0	8.93	4.61
20:0	0.24	0.47
21:0	0.84	0.00
24:0	0.15	0.00
Total saturated fatty acids (SFA)	34.75	43.79
14:1	0.01	0.16
15:1	0.35	0.00
16:1	12.82	7.44
17:1	0.45	0.49
18:1	21.92	18.03
20:1	1.80	3.20
22:1	0.00	0.00
24:1	0.48	0.88
Total monounsaturated fatty acids (MUFA)	37.83	30.2
18:2 ɷ6	15.96	2.65
18:3 ɷ6	0.46	0.00
20:2 ɷ6	0.00	0.00
20:3 ɷ6	0.52	0.00
20:4 ɷ6	4.92	1.45
Total polyunsaturated fatty acids (PUFA) ɷ6	21.86	4.1
18:3 ɷ3	2.95	0.85
18:4 ɷ3	0.26	0.00
20:3 ɷ3	0.00	0.00
20:5 ɷ3 eicosapentaenoic acid (EPA)	1.76	10.08
22:5 ɷ3	0.05	1.97
22:6 ɷ3 docosaesaenoic acid (DHA)	0.54	9.01
Total PUFA ɷ3	5.56	21.91
PUFA ɷ6/PUFA ɷ3	3.93	0.19

**Table 3 animals-09-00144-t003:** Formulation, proximate composition, and essential amino acids of the three diets used in the trial.

Feed	L1	L2	L3
Feedstuffs (g/kg)			
Fish meal	180	135	75
Chironomid meal	0	45	90
Soybean meal	285	300	300
Wheat meal	180	125	125
Hemoglobin	100	100	100
Gluten corn	150	190	205
Fish oil	90	90	90
Vitamin and mineral premix	15	15	15
Chemical composition (%)			
Dry matter	91.19	91.25	91.08
Crude protein	45.47	45.62	45.83
Crude lipid	13.50	13.60	13.94
Ash	5.36	5.14	4.73
Crude fiber	2.12	2.27	2.24
Gross energy (MJ kg)	16.58	16.61	16.73
Essential amino acids (g/kg dry matter)			
Arginine (ARG)	23.5	21.0	19.7
Histidine (HIS)	11.5	10.0	9.5
Isoleucine (ISO)	24.3	22.3	21.5
Leucine (LEU)	34.0	33.0	30.2
Lysine (LYS)	28.5	27.0	26.5
Methionine (MET)	13.0	11.6	10.2
Phenylalanine (PHE)	19.0	24.0	27.5
Threonine (THR)	19.7	24.0	27.3
Tryptophan (TRP)	4.0	4.2	4.6
Valine (VAL)	23.0	26.2	31.4

Feed ingredients were obtained by the Agrarian Corsortium, Sant’Egidio alla Vibrata (TE).

**Table 4 animals-09-00144-t004:** Fatty acid profile of the three diets (% of total fatty acids).

Fatty Acid	L1	L2	L3
14:0	9.89	7.24	5.16
15:0	0.47	0.61	0.53
16:0	24.23	26.72	27.25
17:0	0.54	0.58	0.64
18:0	5.58	5.73	6.28
20:0	0.25	0.13	0.16
Total SFA	40.96	41.01	40.02
14:1	0.11	0.10	0.12
16:1	8.05	10.24	12.12
17:1	0.48	0.45	0.46
18:1	14.23	14.72	14.23
20:1	0.47	0.29	0.26
24:1	0.82	0.80	0.68
Total MUFA	24.16	26.60	27.87
18:2 ɷ6	5.92	10.04	13.02
20:4 ɷ6	1.12	1.37	1.40
Total PUFA ɷ6	7.04	11.41	14.42
18:3 ɷ3	0.59	0.68	0.69
18:4 ɷ3	2.14	2.17	2.45
20:5 ɷ3	10.16	7.04	6.26
22:5 ɷ3	1.35	1.31	1.22
22:6 ɷ3	9.75	5.86	3.10
Total PUFA ɷ3	23.99	17.06	13.72
ɷ6/ɷ3	0.29	0.67	1.05
Others	3.85	3.92	3.97

**Table 5 animals-09-00144-t005:** Growth performances of gilthead sea bream that were fed the three different diets.

**Zootechinical Parameters**	**L1**	**L2**	**L3**
Initial mean body weight (g)	75 ± 1.1	75 ± 1.1	75 ± 1.1
Final mean body weight (g)	174.52 ± 16	170.29 ± 19	171.86 ± 20
Final total mean length (cm)	21.60 ± 1.1	20.04 ± 0.9	21.30 ± 1
Specific growth rate (%)	0.94 ± 0.01	0.91 ± 0.02	0.92 ± 0.03
Survival rate (%)	94.31 ± 2.3	94.13 ± 2.5	93.62 ± 2.2
Feed conversion rate	1.18 ± 0.01	1.21 ± 0.04	1.22 ± 0.04
Palatability	100.0 ± 0.00	100.0 ± 0.00	100.0 ± 0.00
**Morpho-Biometric Parameters and Somatic Indices**	**L1**	**L2**	**L3**
Condition index (KI)	1.76 ± 0.01	1.83 ± 0.02	1.83 ± 0.01
Viscerosomatic index (VSI)	6.42 ± 0.03	6.45 ± 0.05	6.36 ± 0.03
Perivisceral fat index (PFI)	1.03 ± 0.14	1.05 ± 0.12	1.03 ± 0.11
Hepatosomatic index (HSI)	1.42 ± 0.11	1.36 ± 0.14	1.43 ± 0.12

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
