# Peer review of "Growing Trial of Gilthead Sea Bream (Sparus aurata) Juveniles Fed on Chironomid Meal as a Partial Substitution for Fish Meal"

_animals, 2019, doi:10.3390/ani9040144_

Round 1

Reviewer 1 Report

Dear Editor,

The study of meal from Chironomid insects to employ in feed for gilthead sea bream in order to understand the difference in growth is a relevant research topic.

The subject is of interest for the “Animals”.

The experimental design was well conducted and, overall, the document is written in a clear way. Statistics elaboration is suitable to show differences between the two culture systems.

The Discussion puts the results of data analysis into perspective by explaining potential reasons for the observed differences in quality traits and comparing the results to related studies.

Although studies on insect meal are not a novel, information on the nutritional quality of Chironomid insects is very scarce, and it is also foreseen that this species could play an important role in aquafeed.

Therefore, I recommend the acceptance of this manuscript with minor revisions.

Please, find listed these suggestions:

Materials and Methods

-Line 103:

the authors refer that aminoacid (AA) profile of chironomid meal was analysed, but it seems the essential AA were reported only. If available, it should be interesting to wide to all the AA content. For example, if glutamic acid is present in high concentration and compare to AA profile of fish meal and the same in the diets.

Results and Discussion

As consequence to consider the total AA profile of chironomid meal, please give relative results and discuss

Line 224-:

this part of Discussion can be considered alone and implemented as Conclusion, although this section is not mandatory as the Guidelines declare.

Best regards

Author Response

The study of meal from Chironomid insects to employ in feed for gilthead sea bream in order to understand the difference in growth is a relevant research topic. The subject is of interest for the “Animals”. The experimental design was well conducted and, overall, the document is written in a clear way. Statistics elaboration is suitable to show differences between the two culture systems. The Discussion puts the results of data analysis into perspective by explaining potential reasons for the observed differences in quality traits and comparing the results to related studies. Although studies on insect meal are not a novel, information on the nutritional quality of Chironomid insects is very scarce, and it is also foreseen that this species could play an important role in aquafeed. Therefore, I recommend the acceptance of this manuscript with minor revisions.

Response: Thank you so much for your time and effort to review this manuscript, and your high approval of our research.

Please, find listed these suggestions:

Materials and Methods

- Line 103: the authors refer that aminoacid (AA) profile of chironomid meal was analysed, but it seems the essential AA were reported only. If available, it should be interesting to wide to all the AA content. For example, if glutamic acid is present in high concentration and compare to AA profile of fish meal and the same in the diets.

Response: The aminoacid profile was revised with “essential” because only the ten essential aminoacids were analysed.

Results and Discussion

- As consequence to consider the total AA profile of chironomid meal, please give relative results and discuss

Response: we tried to show the essential AA, we think to reply trials extending the analysis to the aminoacid profile.

- Line 224-: this part of Discussion can be considered alone and implemented as Conclusion, although this section is not mandatory as the Guidelines declare.

Response: maintaining the results focalized on the essential AA, this section is left without separation between Discussion and Conclusion.

Reviewer 2 Report

This manuscript describes the results of a feeding experiment with gilthead sea bream which was offered a feed in which part of the fish meal was replaced by chironomid larvae meal. Chironomids are abundant in the benthic fauna of freshwater and brackish environments and form a substantial proportion of the natural feed of many fish species in freshwater or euryhaline habitats. The use of chironomids for fish feed has been addressed in a few previous communications, but there are still many open questions which need to be addressed. In addition, this is the first communication, which considers chironomid meal as a component of the diet of the euryhaline fish species gilthead sea bream, a major species in the Mediterranean aquaculture industry. Hence, the topic of the manuscript is of significance for the field.  The concept of the study is valid, and the results are presented in an acceptable form. There are however some points, which should be addressed before final acceptance.

Lines 26/27 ff. Please consider whether the experimental feed hat the same (or similar) energy content.

Lines 27 ff: and lines 83 ff: the description of the experimental diets could be shortened, just stating that the amount of fish meal was partially replaced by chironomid meal and a reference to table 3, which shows the composition of the diets. In table 3 the suppliers of the feed ingredients are not mentioned and tis should be added.

Feeding trail: it is said that the feed was supplied to the fish ad libitum. Was uneaten feed collected from the tanks and subtracted from the amount of feed given?  The palatability was calculated to 100 % (table 5). This means all the feed given was ingested by the fish? Were then the fish fed until apparent satiation?

Bio-morphometric parameters and indices: Was the total length of the fish determined (including tail fin) or standard length (excluding tail fin)? How many fish were used for calculating VSI, HIS and PSI? Were the fish for this dissected at the end of the feeding experiment? Please add to the manuscript.

Statistical analysis: Was the data tested for normality and homoscedasticity? Please add to the manuscript, if done

Results, line 138: ammonia nitrogen and nitrite: it was probably not absent from tank water, but the concentration was below the detection limit of the test used. Please change to: not detected, or: below detection limit.

Abbreviations in table 1 and 2 and later in the text: please spell out when first mentioned and/ or include a list of abbreviations.

Discussion,

Line 202 ff: This sentence is unclear. Change to: The chironomid meal had a similar composition than the chironomids grown in controlled conditions on pure culture of microalgae (Habib et al. [25]). However, the preparation used in the present study had higher protein and lower lipid contents relative to the chironomids grown in an algal culture by Habib et al. (25).

Line 207: Sentence is strange. What is a “good level”?  Please change to: “The level of tryptophan, leucine, lysine, phenylalanine, threonine and valine was similar to fish meal and placed these species of insects among those with an adequate composition among the Diptera, ….”

Lin 220: This sentence is not clear: What does it mean, that your results are  “in line” with the results from the study cited in this sentence: In the present study, there was no difference in performance between the feeding groups. Your data shows, that the addition of the chronomid meal did not suppress growth performance. Please re-write.

Generally, the manuscript is well written, but needs language editing by a native English speaker. In the text some points are highlighted, but there are many more.

Author Response

This manuscript describes the results of a feeding experiment with gilthead sea bream which was offered a feed in which part of the fish meal was replaced by chironomid larvae meal. Chironomids are abundant in the benthic fauna of freshwater and brackish environments and form a substantial proportion of the natural feed of many fish species in freshwater or euryhaline habitats. The use of chironomids for fish feed has been addressed in a few previous communications, but there are still many open questions which need to be addressed. In addition, this is the first communication, which considers chironomid meal as a component of the diet of the euryhaline fish species gilthead sea bream, a major species in the Mediterranean aquaculture industry. Hence, the topic of the manuscript is of significance for the field.  The concept of the study is valid, and the results are presented in an acceptable form. There are however some points, which should be addressed before final acceptance.

Response: Thank you very much for your time and effort to review this manuscript. This is very helpful to improve our research.

- Lines 26/27 ff. Please consider whether the experimental feed had the same (or similar) energy content.

Response: the energy content of feeds was considered in Materials and Methods.

- Lines 27 ff: and lines 83 ff: the description of the experimental diets could be shortened, just stating that the amount of fish meal was partially replaced by chironomid meal and a reference to table 3, which shows the composition of the diets. In table 3 the suppliers of the feed ingredients are not mentioned and tis should be added.

Response: The sentences were shortened and reference of Table 3 was given. The supplier of feed ingredient was reported in Table 3.

- Feeding trail: it is said that the feed was supplied to the fish ad libitum. Was uneaten feed collected from the tanks and subtracted from the amount of feed given? The palatability was calculated to 100 % (table 5). This means all the feed given was ingested by the fish? Were then the fish fed until apparent satiation?

Response: concerning uneaten feed, the explanation about it was given and the feeding was performed until the apparent satiation.

- Bio-morphometric parameters and indices: Was the total length of the fish determined (including tail fin) or standard length (excluding tail fin)? How many fish were used for calculating VSI, HIS and PSI? Were the fish for this dissected at the end of the feeding experiment? Please add to the manuscript.

Response: Information about total length and morphometric parameters were added in 2.3 paragraph.

- Statistical analysis: Was the data tested for normality and homoscedasticity? Please add to the manuscript, if done

Response: No, data was not tested for normality and homoscedasticity.

- Results, line 138: ammonia nitrogen and nitrite: it was probably not absent from tank water, but the concentration was below the detection limit of the test used. Please change to: not detected, or: below detection limit.

Response: the sentence was corrected with “below detection limit”.

- Abbreviations in table 1 and 2 and later in the text: please spell out when first mentioned and/ or include a list of abbreviations.

Response: abbreviations were reported in Tables and used in the text.

- Line 202 ff: This sentence is unclear. Change to: The chironomid meal had a similar composition than the chironomids grown in controlled conditions on pure culture of microalgae (Habib et al. [25]). However, the preparation used in the present study had higher protein and lower lipid contents relative to the chironomids grown in an algal culture by Habib et al. (25).

Response: the sentence has been changed.

- Line 207: Sentence is strange. What is a “good level”? Please change to: “The level of tryptophan, leucine, lysine, phenylalanine, threonine and valine was similar to fish meal and placed these species of insects among those with an adequate composition among the Diptera, ….”

Response: the sentence has been changed.

- Lin 220: This sentence is not clear: What does it mean, that your results are “in line” with the results from the study cited in this sentence: In the present study, there was no difference in performance between the feeding groups. Your data shows, that the addition of the chronomid meal did not suppress growth performance. Please re-write.

Response: the sentence was re-written.

Generally, the manuscript is well written, but needs language editing by a native English speaker. In the text some points are highlighted, but there are many more.

Response: a colleague mothertongue revised the grammar and style of the text.